# Deep-Trim: Revisiting L1 Regularization for Connection Pruning of Deep Network

## Abstract

State-of-the-art deep neural networks (DNNs) typically have tens of millions of parameters, which might not fit into the upper levels of the memory hierarchy, thus increasing the inference time and energy consumption significantly, and prohibiting their use on edge devices such as mobile phones. The compression of DNN models has therefore become an active area of research recently, with *connection pruning* emerging as one of the most successful strategies. A very natural approach is to prune connections of DNNs via $\ell_1$ regularization, but recent empirical investigations have suggested that this does not work as well in the context of DNN compression. In this work, we revisit this simple strategy and analyze it rigorously, to show that: (a) any *stationary point* of an $\ell_1$-regularized layerwise-pruning objective has its number of non-zero elements bounded by the number of penalized prediction logits, regardless of the strength of the regularization; (b) successful pruning highly relies on an accurate optimization solver, and there is a trade-off between compression speed and distortion of prediction accuracy, controlled by the strength of regularization. Our theoretical results thus suggest that $\ell_1$ pruning could be successful provided we use an accurate optimization solver. We corroborate this in our experiments, where we show that simple $\ell_1$ regularization with an Adamax-L1(cumulative) solver gives pruning ratio competitive to the state-of-the-art.

## 1 Introduction

State-of-the-art Deep Neural Networks (DNNs) typically have millions of parameters. For example, the VGG-16 network (Simonyan and Zisserman (2014)), from the winning team of ILSVRC-2014, contains more than one hundred million parameters; inference with this network on a single image takes tens of billions of operations, prohibiting its use on edge devices such as mobile phones or in real-time applications. In addition, the huge size of DNNs often precludes them from being placed at the upper level of the memory hierarchy, with resulting slow access times and expensive energy consumption.

A recent thread of research has thus focused on the question of how to compress DNNs. One successful approach that has emerged is to trim the connections between neurons, which reduces the number of non-zero parameters and thus the model size (Han et al. (2015a;b); Guo et al. (2016); Molchanov et al. (2017); Aghasi et al. (2017); Dong et al. (2017); Tung and Mori (2018)). However, there has been a gap between the theory and practice: the trimming algorithms that have been practically successful (Han et al. (2015a;b); Guo et al. (2016)) do not have theoretical guarantees, while theoretically-motivated approaches have been less competitive compared to the heuristics-based approaches Aghasi et al. (2017), and often relies on stringent distributional assumption such as Gaussian-distributed matrices which might not hold in practice. With a better theoretical understanding, we might be able to answer how much pruning one can achieve via different approaches on different tasks, and moreover when a given pruning approach might or might not work. Indeed, as we discuss in our experiments, even the generally practically successful approaches are subject to certain failure cases. Beyond simple connection pruning, there have been other works on structured pruning that prune a whole filter, whole row, or whole column at a time (Anwar et al. (2017); Alvarez and Salzmann (2016); Wen et al. (2016); Li et al. (2016); Luo and Wu (2017)). The structured pruning strategy can often speed up inference speed at prediction time more than simple connection pruning,

but the pruning ratios are typically not as high as non-structured connection pruning; so that the storage complexity is still too high, so that the caveats we noted earlier largely remain.

A very natural strategy is to use $\ell_1$ regularized training to prune DNNs, due to their considerable practical success in general sparse estimation in shallow model settings. However, many recent investigations seemed to suggest that such $\ell_1$ regularization does not work as well with non-shallow DNNs, especially compared to other proposed methods. Does $\ell_1$ regularization not work as well in non-shallow models? In this work, we theoretically analyze this question, revisit the trimming of DNNs through $\ell_1$ regularization. Our analysis provides two interesting findings: (a) for any stationary point under $\ell_1$ regularization, the number of non-zero parameters in each layer of a DNN is bounded by the number of penalized prediction logits—an upper bound typically several orders of magnitude smaller than the total number of DNN parameters, and (b) it is critical to employ an $\ell_1$-friendly optimization solver with a *high precision* in order to find the *stationary point of sparse support*.

Our theoretical findings thus suggest that one could achieve high pruning ratios even via $\ell_1$ regularization provided one uses high-precision solvers (which we emphasize are typically not required if we only care about prediction error rather than sparsity). We corroborate these findings in our experiments, where we show that solving the $\ell_1$-regularized objective by the combination of SGD pretraining and Adamax-L1(cumulative) yields competitive pruning results compared to the state-of-the-art.

## 2 PROBLEM FORMULATION

Let $X^{(0)} : N \times D_1^{(0)} \cdots \times D_p^{(0)} \times K^0$ be an input tensor where $N$ is the number of samples (or batch size). We are interested in DNNs of the form

$$X^{(j)} := \sigma_{W^{(j)}}(X^{(j-1)}), \;\; l = j \ldots J$$

where $\sigma_{W^{(j)}}(X^{(j-1)})$ are piecewise-linear functions of both the parameter tensor $W^{(j)} : K^{(j-1)} \times C_0^{(j)} \cdots \times C_p^{(j)} \times K^{(j)}$ and the input tensor $X^{(j-1)} : N \times D_1^{(j-1)} \cdots \times D_p^{(j-1)} \times K^{(j-1)}$ of $(j)$-th layer. Examples of such piecewise-linear function include:

(a) convolution layers with Relu activation (using $\circ$ to denote the $p$-dimensional convolution operator)

$$[\sigma_W(X)]_{i,k} := [\sum_{m=1}^{K^{(j-1)}} X_{i,:,m} \circ W_{m,:,k}]_+,$$

(b) fully-connected layers with Relu activation

$$[\sigma_W(X)]_{i,k} := [X_{i,:}W_{:,k}]_+,$$

(c) commonly used operations such as *max-pooling*, *zero-padding* and *reshaping*.

Note $X^{(J)} : N \times K$ provide $K$ scores (i.e. logits) of each sample that relate to the labels of our target task $Y : N \times K$. Denote $L(X^{(J)}, Y)$ as the task-specific loss function. We define *Support Labels* of a DNN $X^{(J)}$ as indices $(i, k)$ of non-zero loss subgradient w.r.t. the prediction logit:

**Definition 1** (Support Labels). *Let $L(X, Y)$ be a convex loss function w.r.t. the prediction logits $X$. The* Support Labels *regarding DNN outputs $X^{(J)}(W)$ are defined as*

$$\mathcal{S}(W) := \left\{ (i, k) \in [N] \times [K] \;\mid\; [Q]_{i,k} \neq 0, \; \text{for some } Q \in \partial_X L(X^{(J)}, Y) \right\}.$$

*We will denote $k_{\mathcal{S}}(W) := \frac{|\mathcal{S}(W)|}{N} \leq K$ as the average number of* support labels *per sample.*

We illustrate these concepts in the context of some standard machine learning tasks.

**Multiple Regression.** In multiple regression, we are interested in multiple real-valued labels, such as the location and orientation of objects in an image, which over the set of $N$ samples, can be expressed as an $N \times K$ real-valued matrix $Y$. A popular loss function for such tasks is:

$$L(X, Y) := \frac{1}{2} \|X - Y\|_F^2,$$

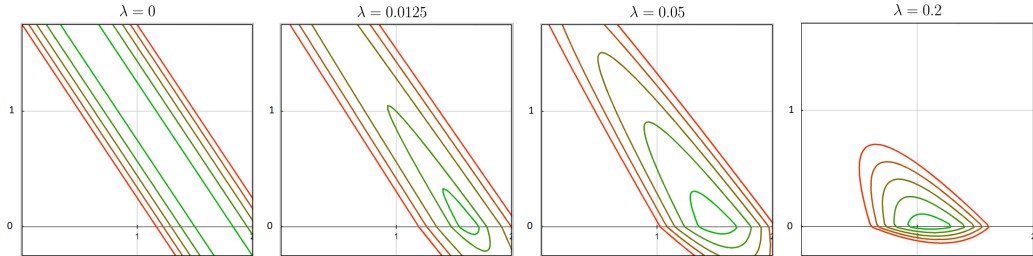

Figure 1: The level curves of the $\ell_1$-regularized objective with 2 parameters and 1 sample: $\lambda\|\boldsymbol{w}\|_1 + \frac{1}{2}(\boldsymbol{x}^T\boldsymbol{w} - 1)^2$ where $\boldsymbol{x} = [0.6, 0.4]$. Note that the stationary points have $w_2 = 0$ as long as $\lambda > 0$. However, smaller $\lambda$ makes convergence to the stationary point harder for optimization algorithms.

which is convex and differentiable, and in general we have $[\nabla_X L]_{i,k} \neq 0$, therefore all labels are support labels (i.e. $k_{\mathcal{S}} = K$).

**Binary Classification.** In binary classification, the labels are binary-valued, and over the set of $N$ samples, can be represented as a binary vector $\boldsymbol{y} \in \{-1, 1\}^N$. Popular loss functions include the *logistic loss*:

$$L(\boldsymbol{x}, \boldsymbol{y}) := \sum_{i=1}^{N} \log(1 + \exp(-y_i x_i)), \tag{1}$$

and the *hinge loss*:

$$L(\boldsymbol{x}, \boldsymbol{y}) := \sum_{i=1}^{N} [1 - y_i x_i]_+. \tag{2}$$

For the logistic loss (1), we have $k_{\mathcal{S}} = 1$ since $[\nabla L]_i \neq 0, \forall i \in [N]$. On the other hand, the hinge loss (2) typically has only a small portion of samples $i \in [N]$ with $[\partial L]_i \neq \{0\}$, called *Support Vectors*, and which coincides with our definition of *Support Labels* in this context. In applications with unbalanced positive and negative examples, such as *object detection*, we have $k_{\mathcal{S}} \ll 1$.

**Multiclass/Multilabel Classification.** In multiclass or multilabel classification, the labels of each sample can be represented as a $K$-dimensional binary vector $\{0, 1\}^K$ where 1/0 denotes the presence/absence of a class in the sample. Let $\mathcal{P}_i := \{k \mid y_{ik} = 1\}$ and $\mathcal{N}_i := \{k \mid y_{ik} = 0\}$ denote the positive and negative label sets. Popular loss functions include the cross-entropy loss:

$$L(X, Y) := \sum_{i=1}^{N} \left( \log \sum_{k=1}^{K} \exp(X_{ik}) - \frac{1}{|\mathcal{P}_i|} \sum_{k \in \mathcal{P}_i} X_{ik} \right), \tag{3}$$

and the maximum margin loss:

$$L(X, Y) := \sum_{i=1}^{N} \left( \max_{j \in \mathcal{N}_i, k \in \mathcal{P}_i} 1 + X_{ij} - X_{ik} \right). \tag{4}$$

Although the cross-entropy loss (3) has number of support labels $k_{\mathcal{S}} = K$, it has been shown that the maximum-margin loss (4) typically has $k_{\mathcal{S}} \ll K$ in recent studies of classification problems with extremely large number of classes (Yen et al. (2016; 2017)).

## 3 DEEP-TRIM: THEORY

In this section, we aim to solve the following DNN compression problem.

**Definition 2** (Deep-Trim ($\epsilon$))**.** *Suppose we are given a target loss function $L(X, Y)$ between prediction $X$ and training labels $Y : N \times K$, and a pre-trained DNN $X^{(J)}$ parameterized by weights $W := \{W^{(j)}\}_{j=1}^{J}$, with loss $L^* := L(X^{(J)}(W), Y)$. The Deep-Trim ($\epsilon$) task is to find a compressed DNN with weights $\widehat{W}$ such that its number of non-zero parameters $nnz(\widehat{W}) \leq \tau$, for some $\tau \ll nnz(W)$ and where $L(X^{(J)}(\widehat{W}), Y) \leq L^* + \epsilon$ for some $\epsilon > 0$.*

In the following, we show that the *Deep-Trim* problem with budget $\tau = (N k_{\mathcal{S}}) \times J$ can be solved via simple $\ell_1$ regularization under a couple of mild conditions, with the caveat that with suitable optimization algorithms be used, and where $k_{\mathcal{S}}$ is the maximum number of *support labels* for any $W$ with $L(X^{(L)}(W), Y) \leq L^*$.

**Trimming Objective**   Given a loss function $L(., Y)$ and a pre-trained DNN parameterized by $W^* := \{W^{(j)}\}_{j=1}^J$, we initialize the iterate with $W^*$ and apply an optimization algorithm that *guarantees descent* of the following layerwise $\ell_1$-regularized objective

$$\min_{W^{(j)}} \left\{ \lambda \|vec(W^{(j)})\|_1 + L(X^{(J)}(W), Y) \right\}, \tag{5}$$

for all $j \in [J]$, where $vec(W^{(j)})$ denotes the vectorized version of the tensor $W^{(j)}$.

The following theorem states that most of stationary points of (5) have the number of non-zero parameters per layer bounded by the total number of support labels in the training set.

**Theorem 1** (Deep-Trim with $\ell_1$ penalty). *Let $\widehat{W}^{(j)}$ be any stationary point of objective (5) with $dim(W^{(j)})=d$ that lies on a single linear piece of the piecewise-linear function $X^{(J)}(W)$. Let $V : (NK) \times d$ be the Jacobian matrix of that corresponding linear piece of the linear (vector-valued) function $vec(X^{(J)})(vec(W^{(j)}))$. For any regularization parameter $\lambda > 0$ and $V$ in general position we have*

$$nnz(\widehat{W}^{(j)}) \leq N\, k_\mathcal{S}(\hat{W}).$$

*where $k_\mathcal{S}(\widehat{W})$ is the average number of support labels of the stationary point $\widehat{W}^{(j)}$.*

*Proof.* Any stationary point of (5) should satisfy the condition

$$V^T vec(A) + \lambda \boldsymbol{\rho} = 0 \tag{6}$$

where $A \in \partial L$ is an $N \times K$ subgradient matrix of the loss function w.r.t. the prediction logits, and $\boldsymbol{\rho} \in \partial \|vec(W^{(j)})\|_1$ is a $d$-dimensional subgradient of the $\ell_1$ norm penalty. Then let $\mathcal{Q} := \{r \mid [vec(W^{(j)})]_r \neq 0\}$ be the set of indices of non-zero parameters, we have $[\boldsymbol{\rho}]_r \in \{-1, 1\}$ and thus the linear system

$$\left[ V^T vec(A) \right]_\mathcal{Q} = -\lambda [\boldsymbol{\rho}]_\mathcal{Q}, \tag{7}$$

cannot be satisfied if $nnz(A) < nnz(W^{(j)})$ for $V$ is in general position (as defined in, for example, Tibshirani et al. (2013)). Therefore, we have $nnz(A) = Nk_\mathcal{S} \geq nnz(W^{(j)}) = |\mathcal{Q}|$.   □

Note the concept of *general position* is studied widely in the literature of LASSO and sparse recovery, and it is a weak assumption in the sense that any matrix drawn from a continuous probability distribution is in *general position* (Tibshirani et al. (2013)).

Figure 1 illustrates an example of a regression task where, no matter how small $\lambda > 0$ is, the second coordinate is always 0 at the stationary point. Note since Theorem 1 holds for any $\lambda > 0$, one can guarantee to trim a DNN without hurting the training loss by choosing an appropriately small $\lambda$, as stated by the following corollary.

**Corollary 1** (Deep-Trim without Distortion). *Given a DNN with weights $W$ and with loss $L^* := L(X^{(J)}(W), Y)$, for any $\epsilon > 0$, one can find weights $\widehat{W}$ with $L(X^{(J)}(\widehat{W}), Y) \leq L^* + \epsilon$, and $nnz(\widehat{W}^{(j)}) \leq J\,N\,k_\mathcal{S}$, where $k_\mathcal{S}$ is a bound on the number of* support labels *of parameters $\widehat{W}$ with loss no more than $L^* + \epsilon$.*

*Proof.* By choosing $\lambda \leq \epsilon/(J\|vec(W^{(j)})\|_1)$, any descent optimization algorithm can guarantee to find $\hat{W}^{(j)}$ with

$$\lambda \|vec(\hat{W}^{(j)})\|_1 + L(X^{(J)}(W^{(1)}, \ldots, \hat{W}^{(j)}, \ldots, W^{(J)}), Y) \leq \lambda \|vec(W^{(j)})\|_1 + L^*$$

by minimizing (5), which guarantees $L(X^{(J)}(W^{(1)}, \ldots, \hat{W}^{(j)}, \ldots, W^{(J)}), Y) \leq L^* + \epsilon/J$ and $nnz(W^{(j)}) \leq Nk_\mathcal{S}$. Then by applying the procedure for each layer $j \in [J]$, one can obtain $\hat{W}$ with $L(X^{(J)}(\hat{W}), Y) \leq L^* + \epsilon$ and $nnz(\hat{W}) \leq (Nk_\mathcal{S}) \times J$.   □

In practice, however, the smaller $\lambda$, the harder for the optimization algorithm to get close to the stationary point, as illustrated in the figure 1. Therefore, it is crucial to choose optimization algorithms targeting for *high precision* for the convergence to the stationary point of (5) with sparse support, while the widely-used Stochastic Gradient Descent (SGD) method is notorious for being inaccurate in terms of the optimization precision.

## 4 DEEP-TRIM: ALGORITHMS

Although our analysis is conducted on the layerwise pruning objective (5), in practice we have observed joint pruning of all layers to be as effective as layerwise pruning. For ease of presentation of this section, we will denote our objective function

$$\min_{vec(W)\in\mathbb{R}^d} \lambda\|vec(W)\|_1 + L(X^{(J)}(W), Y) \tag{8}$$

in the following form

$$\min_{\boldsymbol{w}\in\mathbb{R}^d} \lambda\|\boldsymbol{w}\|_1 + f(\boldsymbol{w}) \tag{9}$$

where $\boldsymbol{w} := vec(W)$ and $f(\boldsymbol{w}) := L(X^{(J)}(W), Y)$. Note the same formulation (9) can be also used to represent the layerwise pruning objective (5) by simply replacing their definitions as

$$\boldsymbol{w} := vec(W^{(j)}) \text{ and } f(\boldsymbol{w}) := L(X^{(J)}(W^{(1)}, \ldots, W^{(j)}, \ldots, W^{(J)}), Y).$$

As mentioned previously, even when the stationary point of an objective has sparse support, if the optimization algorithm does not converge close enough to the stationary point, the iterates would still have very dense support. In this section, we propose a two-phase strategy for the non-convex optimization problem (9). In the first phase, we initialize with the given model and use a simple Stochastic Gradient Descent (SGD) algorithm to optimize (9). During this phase, we do not aim to reduce the number of non-zero parameters but only to reduce the $\ell_1$ norm of the model parameters. We run the SGD till both the training loss and $\ell_1$ norm of model parameters have converged. Then in the second phase, we employ an Adamax-L1 (cumulative) method to reduce the total number of non-zero parameters, and achieves pruning result on-par with state-of-the-art methods.

**SGD with L1 Penalty**    For a simple optimization problem $\min_{\boldsymbol{w}\in\mathbb{R}^d} f(\boldsymbol{w})$, the SGD update follows the form $w^{t+1} = w^t - \eta_t \frac{\partial f(\boldsymbol{w})}{\partial \boldsymbol{w}}$. We consider general SGD-like algorithms which update in the form $w^{t+1} = w^t - \eta_t g(\frac{\partial f(\boldsymbol{w})}{\partial \boldsymbol{w}}, \theta)$, where $\theta$ is a set of parameters specific to the SGD-like update procedure. This includes the commonly used Momentum (Qian (1999)), Adamax, Adam (Kingma and Ba (2014)), and RMSProp (Tieleman and Hinton (2012)) optimization algorithms.

When employing SGD-like optimizers, (9) can be rewritten as the following:

$$\min_{\boldsymbol{w}\in\mathbb{R}^d} \sum_{j}^{N_b} (f(\boldsymbol{w}, j) + \frac{\lambda}{N}\|\boldsymbol{w}\|_1), \tag{10}$$

where $j$ denotes one mini-batch of data and $N_b$ is the number of mini-batches. The weight updated by the SGD-like optimizers can then be performed as

$$\boldsymbol{w}_i^{t+1} = \boldsymbol{w}_i^t - \eta_t \cdot g(\frac{\partial f(\boldsymbol{w}, j)}{\partial \boldsymbol{w}_i} + \frac{\lambda}{N_b}\text{sign}(\boldsymbol{w}_i^t), \theta), \tag{11}$$

where $\text{sign}(\boldsymbol{w}_i) = 0$ when $\boldsymbol{w}_i = 0$. We note that after the update in (11), the weight does not become 0 unless $\boldsymbol{w}_i^t = \eta_t \cdot g(\frac{\partial f(\boldsymbol{w}, j)}{\partial \boldsymbol{w}_i} + \frac{\lambda}{N_b}\text{sign}(\boldsymbol{w}_i^t))$, which rarely happens. Therefore, adding the L1 penalty term to SGD-like optimizers only minimizes the L1-norm but does not induce a sparse weight matrix. To achieve a sparse solution, we combine the L1 friendly update trick SGD-L1 (cumulative) (Tsuruoka et al. (2009)) along with SGD-like optimization algorithms.

**Adamax-L1 (cumulative)**    SGD-L1 (clipping) is an alternative to perform L1 regularizing along with SGD to obtain a sparse $\boldsymbol{w}$ (Carpenter (2008)). Different to (11), SGD-L1 (clipping) divides the update into two steps. The first step is updated without considering the L1 penalty term, and the second step updates the L1 penalty separately. In the second step, any weight that has changed its sign during the update will be set to 0. In other words, when the L1 penalty is larger than the weight value, it will be truncated to the weight value. Therefore, SGD-L1 (clipping) can be seen as a special case of truncated gradient. With a learning rate $\eta_k$, the update algorithm can be written as

$$\boldsymbol{w}_i^{t+\frac{1}{2}} = \boldsymbol{w}_i^t - \eta_k \cdot g(\frac{\partial f(\boldsymbol{w}, j)}{\partial \boldsymbol{w}_i}, \theta),$$

$$\text{if } \boldsymbol{w}_i^{t+\frac{1}{2}} > 0, \boldsymbol{w}_i^{t+1} = \max(0, \boldsymbol{w}_i^{t+\frac{1}{2}} - \frac{\lambda}{N_b}\eta_t), \tag{12}$$

$$\text{if } \boldsymbol{w}_i^{t+\frac{1}{2}} \leq 0, \boldsymbol{w}_i^{t+1} = \min(0, \boldsymbol{w}_i^{t+\frac{1}{2}} + \frac{\lambda}{N_b}\eta_t).$$

SGD-L1 (cumulative) is a modification of the SGD-L1 (clipping) algorithm proposed by Tsuruoka et al. (2009), but uses the cumulative L1 penalty instead of the standard L1 penalty. The intuition is that the cumulative L1 penalty is the amount of penalty that would be applied on the weight if true gradient is applied instead of stochastic gradient. By applying the cumulative L1 penalty, the weight would not be moved away from zero by the noise of the stochastic gradient. When applied to SGD-like optimization algorithms, the update rule can be written as

$$u_k = \sum_{t=1}^{k} \frac{\lambda}{N_b} \eta_t,$$

$$\boldsymbol{w}_i^{t+\frac{1}{2}} = \boldsymbol{w}_i^t - \eta_k \cdot g(\frac{\partial f(\boldsymbol{w})}{\partial \boldsymbol{w}_i}, \theta),$$

$$\text{if } \boldsymbol{w}_i^{t+\frac{1}{2}} > 0, \boldsymbol{w}_i^{t+1} = \max(0, \boldsymbol{w}_i^{t+\frac{1}{2}} - (u_k + q_i^{k-1})),$$

$$\text{if } \boldsymbol{w}_i^{t+\frac{1}{2}} \leq 0, \boldsymbol{w}_i^{t+1} = \min(0, \boldsymbol{w}_i^{t+\frac{1}{2}} + (u_k - q_i^{k-1})),$$

$$(13)$$

where $q_i^k$ is the total amount of L1 penalty received until now $q_i^k = \sum_{t=1}^{k}(w_i^{t+1} - w_i^{t+\frac{1}{2}})$.

By updating with (13) and adopting the Adamax optimization algorithm (Kingma and Ba (2014)), we obtain Adamax-L1 (cumulative). Originally, SGD-L1 (cumulative) was proposed to be used with the vanilla SGD optimizer, where we generalize it to be used with any SGD-like optimizer by separating the update on objective $f(\boldsymbol{w})$ and the l1-cumulative update on $\lambda\|\boldsymbol{w}\|_1$.

## 5 EXPERIMENTS

In this section, we compare the $\ell$-regularized pruning method discussed in section 4 with other state-of-the-art approaches. In section 5.1, we evaluate different pruning methods on the convolution network *LeNet-5* [1] on the Mnist data set. In section 5.2, we compare our method to VD on pruning *VGG-16* network Simonyan and Zisserman (2014) on the CIFAR-10 data set. In section 5.3, we then conduct experiments with Resnet on CIFAR-10. Finally, we show the trade-off for pruning Resnet-50 on the ILSVRC dataset.

| Method | Acc.% | *nnz* per Layer% | $\frac{\|W\|}{\|W\|_{\neq 0}}$ | FLOP% |
|---|---|---|---|---|
| Prune | 99.2 | $68 - 12 - 8.0 - 19.0$ | 12 | 16 |
| DNS | 99.09 | $14 - 3.1 - 0.7 - 4.3$ | 111 | – |
| VD | 99.25 | $33 - 2.0 - 0.2 - 5.0$ | 280 | – |
| L1 Naive | 99.25 | $100 - 100 - 100 - 100$ | 1 | 100 |
| Ours | 99.05 | $20 - 1.9 - 0.2 - 3.2$ | 260 | 1.2 |

Table 1: Compression Results with LeNet-5 model on MNIST.

### 5.1 LENET-5 ON MNIST

We first compare our methods with other compression methods on the standard MNIST dataset with the LeNet-5 architecture. We consider the following methods: **Prune:** The pruning algorithm proposed in Han et al. (2015a), which iterates between pruning the network after training with L2 regularization and retraining. **DNS:** *Dynamic Network Surgery* pruning algorithm proposed in Guo et al. (2016), which was reported to improve upon the *iterative pruning* method proposed in Han et al. (2015a) by dynamically pruning and splicing variables during the training process. **VD:** *Variational Dropout* method introduced by Molchanov et al. (2017), a variant of dropout that induces sparsity during the training process with unbounded dropout rates. **L1 Naive:** Ablation study of our method by training the $\ell_1$-regularized objective with SGD. **Ours:** Our method which optimizes the $\ell_1$-regularized objective (8) in two phases (SGD and Adamax-L1(cumulative)).

The LeNet-5 network is trained from a random initialization and without data augmentation which achieves 99.2% accuracy. We report the per layer sparsity and the total reduction of weights and

---

[1]We use the modified version of LeNet5 from (LeCun et al., 1998) with Caffe Model specification: `https://goo.gl/4yI3dL`.

| Models | Base Model | | VD | | Ours | |
|---|---|---|---|---|---|---|
| Layers | Weights | FLOP | Weights% | FLOP% | Weights% | FLOPS% |
| Conv1_1 | 1.7K | 716K | 67.8 | 65.8 | 38.4 | 38.4 |
| Conv1_2 | 36.8K | 17.3M | 25.1 | 39.3 | 16.8 | 26.6 |
| Conv2_1 | 73.7K | 11.7M | 33.8 | 28.9 | 27.5 | 28.4 |
| Conv2_2 | 147.5K | 15.5M | 27.6 | 28.3 | 24.3 | 25.2 |
| Conv3_1 | 294.9K | 7.9M | 22.1 | 20.9 | 20.0 | 20.4 |
| Conv3_2 | 589.8K | 8.7M | 11.5 | 11.1 | 11.2 | 11.7 |
| Conv3_3 | 589.8K | 7.8M | 8.6 | 7.8 | 6.0 | 5.1 |
| Conv4_1 | 1.2M | 6.4M | 2.3 | 0.83 | 1.4 | 0.75 |
| Conv4_2 | 2.4M | 8.8M | 0.65 | 0.04 | 0.22 | 0.13 |
| Conv4_3 | 2.4M | 13.4M | 0.23 | 0.03 | 0.11 | 0.11 |
| Conv5_1 | 2.4M | 6.3M | 0.07 | 0.02 | 0.04 | 0.04 |
| Conv5_2 | 2.4M | 5.7M | 0.06 | 0.06 | 0.04 | 0.04 |
| Conv5_3 | 2.4M | 6.7M | 0.05 | 0.04 | 0.07 | 0.07 |
| FC6 | 262.1K | 239.0K | 0.79 | 0.2 | 2.6 | 2.7 |
| FC7 | 5.1K | 1.0K | 41.8 | 57.3 | 88.1 | 75.3 |
| Total | 15.0M | 117.3M | 2.1 | 15.6 | 1.8 | 13.0 |
| $\frac{\|W\|}{\|W\|_{\neq 0}}$ | 1 | | 47.5 | | 57.1 | |
| Acc. % | 92.9 | | 92.2* | | 92.8 | |

Table 2: Compression Results with VGG-like model on CIFAR-10 for VD and our method.

FLOP in Table 1. For LeNet-5, our method achieves comparable sparsity performance against other methods, with a slight accuracy drop. Nevertheless, our compressed model still achieves over 99 percent testing accuracy, while achieving $260\times$ weight reduction and $84\times$ FLOP reduction. We also observe that the *L1 Naive* does not induce any sparsity, even when the L1-norm is significantly reduced. This demonstrates the effectiveness of adopting a L1-friendly optimization algorithm.

## 5.2 VGG-LIKE ON CIFAR-10

To test how our method works on large scale modern architecture, we perform experiments on the VGG-like network with CIFAR-10 dataset, which is used in Molchanov et al. (2017). The network contains 13 convolution layers and 2 fully connected layers and achieves 92.9% accuracy with pretraining. We report the per layer weights and FLOP reduction for our Deep-Trim algorithm and VD (Molchanov et al. (2017)) in Table 5.

Our model achieves a weight pruning ratio of $57\times$ and reduces FLOP by $7.7\times$ with a negligible accuracy drop, and VD achieves $48\times$ weight pruning ratio and reduces FLOP by $6.4\times$.[2] Compared to VD, our model achieved sparser weights from Conv1_1 to Conv5_2 and VD achieved sparser weights from Conv5_2 to FC layers. Interestingly, we observe that in both pruning methods, most remaining *nnz* and FLOPs lie in block2 and block3, where originally block4 and block5 have dominating amount of weights and equal amount of FLOPs.

The layer with the most non-zero parameters after pruning is conv3_2 with $65.9K$. In the experiments we employ the *cross-entropy loss* (3) which has a number of support labels $NK = 500K$ on the CIFAR-10 data set. We suspect a more careful analysis could improve our Theorem 1 to give a tighter bound for loss with entries of gradient close to 0 but not exactly 0, making the bound for *cross-entropy loss* (3) closer to that of *maximum-margin loss* (4).

---

[2]We ran the experiments based on authors' code and tuned the coefficient of dropout regularization loss within the interval $[10^2, 10^{-3}]$ with binary search. We note that although we are able to reproduce the $48\times$ weight reduction ratio in the VD paper, we are only able to achieve Acc. 92.2% instead of 92.7% as reported in their paper.

## 5.3 RESNET-32 ON CIFAR-10

While VGG-network are notorious for its large parameter size, it is not surprising that a large compression rate can be achieved. Therefore, we evaluate the compression performance of our Deep-Trim algorithm on a smaller Resnet-32 model trained on CIFAR-10 data. The Resnet-32 model contains 3 main blocks. The first block contains the first 11 convolution layers with 64 filters in each layer, the second block contains the next 10 convolution layers with 128 filters each, and the last block contains 10 convolution layers with 256 filters and a fully connected layer. We list the detailed architecture in the supplementary. The pretrained Resnet-32 model reaches $94.0\%$ accuracy.

We evaluate our Deep-Trim algorithm and compare it to variational dropout (Molchanov et al. (2017)) and report the results in Table 3. We report the pruning results for each main block of the resnet-32 model. Our model achieves a $33\times$ overall pruning ratio and $21\times$ reduced FLOP with an accuracy drop of $1.4\%$, where VD has attained $28\times$ overall pruning ratio and $13.5\times$ reduction with similar accuracy. We further observe that $nnz$(W) increases much gentler from the first block to the third block compared to the total number of parameters in each block. This is not surprising since the upper bound of $nnz$(W) per layer given by Corollary 1 does not depend on the total number of unpruned parameters.

| Method | Acc.% | $nnz$(W) per block | $\frac{\|W\|}{\|W\|_{\neq 0}}$ | FLOP% |
|---|---|---|---|---|
| Resnet Ref. | 94.0 | 371K-1.4M-5.6M | 1 | 100 |
| VD | 92.5 | 37.8K-90.4K-136.9K | 27.6 | 7.4 |
| Ours | 92.6 | 34.6K-65.2K-120.9K | 33.3 | 4.8 |

Table 3: Compression Results with Resnet-32 model on CIFAR-10.

## 5.4 EFFECT OF DATA ON PRUNING

In this section, we compare the pruning results of our method on VGG-16 with different number of samples. The pruning ratio and number of non-zero parameters are shown in Table 4, we can see that the number of non-zero parameters after pruning clearly grows with the number of samples. This can be understood intuitively, as the number of constraints to be satisfied grows in the training set, the more degree of freedom the model needs to fit the data. This shows that our theory analysis matches our empirical results well.

| Training data | Train Acc.% | Test Acc.% | $nnz$(W) | $\frac{\|W\|}{\|W\|_{\neq 0}}$ |
|---|---|---|---|---|
| 50,000 | 100.0 | 92.8 | 262.1K | 57.1 |
| 5,000 | 100.0 | 84.7 | 76.3K | 196.3 |
| 500 | 100.0 | 57.9 | 18.9K | 792.5 |

Table 4: Compression Results with VGG-like on CIFAR-10 with varying number of training data.

## 6 CONCLUSION

In this work, we revisit the simple idea of pruning connections of DNNs through $\ell_1$ regularization. While recent empirical investigations suggested that this might not necessarily achieve high sparsity levels in the context of DNNs, we provide a rigorous theoretical analysis that does provide small upper bounds on the number of non-zero elements, but with the caveat that one needs to use a high-precision optimization solver (which is typically not needed if we care only about prediction error rather than sparsity). When using such an accurate optimization solver, we can converge closer to stationary points than traditional SGD, and achieve much better pruning ratios than SGD, which might explain the poorer performance of $\ell_1$ regularization in recent investigations. We perform experiments across different datasets and networks and demonstrate state-of-the-art result with such simple $\ell_1$ regularization.

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

# SUPPLEMENTARY MATERIAL

## A   DETAILED ARCHITECTURE FOR RESNET-32

| Layers | Input | number of channels/output units |
|--------|-------|:-------------------------------:|
| Conv1_0 | 32*32 | 64 |
| Conv1_1 | 32*32 | 64 |
| Conv1_2 | 32*32 | 64 |
| Conv1_3 | 32*32 | 64 |
| Conv1_4 | 32*32 | 64 |
| Conv1_5 | 32*32 | 64 |
| Conv1_6 | 32*32 | 64 |
| Conv1_7 | 32*32 | 64 |
| Conv1_8 | 32*32 | 64 |
| Conv1_9 | 32*32 | 64 |
| Conv1_10 | 32*32 | 64 |
| Conv2_1 | 16*16 | 128 |
| Conv2_2 | 16*16 | 128 |
| Conv2_3 | 16*16 | 128 |
| Conv2_4 | 16*16 | 128 |
| Conv2_5 | 16*16 | 128 |
| Conv2_6 | 16*16 | 128 |
| Conv2_7 | 16*16 | 128 |
| Conv2_8 | 16*16 | 128 |
| Conv2_9 | 16*16 | 128 |
| Conv2_10 | 16*16 | 128 |
| Conv3_1 | 8*8 | 256 |
| Conv3_2 | 8*8 | 256 |
| Conv3_3 | 8*8 | 256 |
| Conv3_4 | 8*8 | 256 |
| Conv3_5 | 8*8 | 256 |
| Conv3_6 | 8*8 | 256 |
| Conv3_7 | 8*8 | 256 |
| Conv3_8 | 8*8 | 256 |
| Conv3_9 | 8*8 | 256 |
| Conv3_10 | 8*8 | 256 |
| FC3_11 | 256 | 10 |

Table 5: Per-layer Resnet-32 architecture. There are 3 main convolutional blocks with downsampling through stride=2 for the first layer of each block. After the convloutional layers, global pooling is applied on the spatial axes and a fully-connected layer is appended for the output. Each set of rows is a residual block.

