# OpenReview forum: "DEEP-TRIM: REVISITING L1 REGULARIZATION FOR CONNECTION PRUNING OF DEEP NETWORK"
_ICLR.cc/2019/Conference_

### Official Review · AnonReviewer2 · 2018-10-31
**Nice Theoretical Insights, but Not Sure How Experiments  Substantiate the Theory**

**Rating:** 4
**Confidence:** 3

**Review:**

The paper theoretically analyzes the sparsity property of the stationary point of layerwise l1-regularized network trimming. Experiments are conducted to show that reaching a stationary point of the optimization can help to deliver good performance. Specific comments follow.

1. While the paper analyzes the properties of the stationary point of the layerwise objective (5), the experiments seem to be conducted based on the different joint objective (8). Experimental results of optimizing (5) seem missing. While the reviewer understands that (5) and (8)  are closely related, and the theoretical insights for (5) can potentially translate to the scenario in (8), the reviewer is not sure whether the theory for (5)  is rigorously justified by the experiments.

2. It is also unclear how tight the bound provided by Theorem 1 is.  Is the bound vacuous? Relevant statistics in the experiments might need to be reported to elucidate this point.

3. It is also unclear how the trade-off in point (b) of the abstract is justified in the experiments.

Minor Points:
page 2, the definition of $X^{(j)}$, the index of $l$ and $j$ seem to be typos.
page 2, definition 1, the definition of the bracket need to be specified.
page 4, the concept of stationary point and general position can be introduced before presenting Theorem 1 to improve readability.
page 4, Corollary 1, should it be $nnz(\hat{W})\le JN k_{\mathcal{S}}$?
page 7, Table 2, FLOPS should be FLOP?
page 8, is FLOP related to the time/speed needed for compression? If so, it should be specified. If not, compression runtime should also be reported.

---

> ### Author Response · Authors · 2018-11-26
> **Reply to AnonReviewer2**
>
> We thank the reviewer for the feedback and comments.
>
> (1) "whether the theory for (5)  is rigorously justified by the experiments":
>
> While our theorem is designed for the layerwise objective (5), in practice for simplicity we find that directly optimize (8) yields promising results is more simple. We will show experimental results for both (5) and (8) in future revisions. Note that by optimizing (8), we achieve satisfactory results satisfying our bounds from analyzing (5) in all experiments in this work.
>
> (2) Regarding the bound tightness:
>
> We perform experiments on Cifar 10 with Vgglike-networks with different \lambda values by compressing the last 2 FC layer.
> We would like to point out that the bound for NNZ per-layer in this setting is 50000 * K_s, which depends on the number of supports in the stationary point.
>
> If a max-margin loss is used, K_s can be close to 1, which would give us an NNZ bound around 50000, which is not far from the empirical compressed NNZ (~ 10000).
>
> epsilon     |   1e-4     |    1e-5   |   1e-6    |   1e-7    |  1e-8   |   1e-9  | 1e-10  |      0     |
> nnz_fc1    |   9052    |    9947   |   10046 |   10053 | 10054  | 10054 | 10054 | 262144|
> nnz_fc2    |   4549    |    4567   |   4570   |   4570   |  4570   |  4570  |  4570  |   5120  |
> train_acc  |  0.9970  |  0.9974  |  0.9979 |  0.9970 | 0.9969 | 0.9972| 0.9969| 0.9970 |
>
> (3) regarding minor points:
>
> We will fix the mistakes and typos in future revisions.

---

### Official Review · AnonReviewer1 · 2018-11-04
**an interesting perspective on the L1 regularization of neural network**

**Rating:** 6
**Confidence:** 3

**Review:**

This paper discusses the effect of L1 penalization for deep neural network. In particular it shows the stationary point of an l1 regularized layer has bounded non-zero elements.

The perspective of the proof is interesting: By chain rule, the stationary point satisfies nnz(W^j) linear equations, but the subgradients of the loss function w.r.t. the logits have at most N\times ks variables. If the coefficients of the linear equation are distributed in general positions, then the number of variables should not be larger than the number of equations.

While I mostly like the paper, I would like to point out some possible issues:

main concerns:

1. the columns of V may not be independent during the optimization(training) process. In this situation, I am not quite sure if the assumption of “general position” still holds. I understand that in literatures of Lasso and sparse coding it is common to assume “general position”. But in those problems the coefficient matrix is not Jacobian from a learning procedure.

2. the claim is a little bit counter intuitive: Theorem 1 claims the sparse inequality holds for any \lambda. It is against the empirical observation that when lambda is extremely small, effect of the regularizer tends to be almost zero. Can authors also show this effects empirically, i.e., when the regularization coefficients decrease, the nnz does not vary much? (Maybe there is some optimization details or approximations I missed?)

Some minor notation issues:
1. in theorem 1: dim(W^{(j)})=d should be dim(vec(W^{(j)}))=d
2. in theorem 1: Even though I understand what you are trying to say, I would suggest we describe the jacobian matrix V in details. Especially it is confusing to stack vec(X^J) (vec(W^j)) in the description.
3. the notations of subgradient and gradient are used without claim

---

> ### Author Response · Authors · 2018-11-26
> **Reply to AnonReviewer1**
>
> We thank the reviewer for the nice feedback and concerns.
>
> (1) the assumption of “general position”:
>
> The columns of V do not need to be independent to be in general position. It is sufficient if V is drawn from any continuous probability distribution. In other words, the assumption holds as long as we add a very small continuously-distributed perturbation to V. Note general position is a much weaker condition than the RIP condition used widely in sparse recovery.
>
>
> (2) Theorem 1 claims the sparse inequality holds for any \lambda:
>
> To validate that the sparse inequality holds for any \lambda, we perform experiments on Cifar 10 with Vgglike-networks with different \lambda values by compressing the last 2 FC layer.
> The result is shown below:
>
> epsilon     |   1e-4     |    1e-5   |   1e-6    |   1e-7    |  1e-8   |   1e-9  | 1e-10  |      0     |
> nnz_fc1    |   9052    |    9947   |   10046 |   10053 | 10054  | 10054 | 10054 | 262144|
> nnz_fc2    |   4549    |    4567   |   4570   |   4570   |  4570   |  4570  |  4570  |   5120  |
> train_acc  |  0.9970  |  0.9974  |  0.9979 |  0.9970 | 0.9969 | 0.9972| 0.9969| 0.9970 |
> test_acc   |  0.9271  |  0.9270  |  0.9266 |  0.9267 | 0.9264 | 0.9262| 0.9265| 0.9268 |
>
> We note that we perform SGD with L1 regularizer to train the network as a pretraining step. Empirically, we find that after the L1 norm is penalized, even a very small epsilon can lead to very sparse solutions. (However, when epsilon is too small, the converging time may grow a lot.) For epsilon >= 1e-9, the nnz_fc1 becomes <= 10054 for the first training epoch. However, for epsilon = 1e-10, nnz_fc1 drops to 10054 after the second epoch.
>
> (2) regarding minor points:
>
> We will fix the mistakes and typos in future revisions.

---

### Official Review · AnonReviewer4 · 2018-11-07
**Repeating the old story from other papers, quit limited novelty, lacking solid experiments**

**Rating:** 4
**Confidence:** 4

**Review:**

The main concerns come from the following parts:


(1) Repeating the old story from other papers:
A large part of math is from previous works, which seems not enough for the ICLR conference.
It is very surprising that the authors totally ignore the latest improvements in neural network compression. Their approach is extremely far away from the state of the art in terms of both methodological excellence and experimental results. The authors should read through at least some of the papers I list below, differentiate their approach from these pioneer works, and properly justify their position within the literature. They also need to show a clear improvement on all these existing pieces of work.

(2) quite limited novelty:
In my opinion, the core contribution is replacing SGD with Adam.
For network compression, it is common to add L1 Penalty to loss function. The main difference of this paper is change SGD to Adam, which seems not enough.

(3) lacking solid experiments:
In section Experiment, the authors claim "Finally, we show the trade-off for pruning Resnet-50 on the ILSVRC dataset.", but I cannot find the results.

Is the ResNet-32 too complex for cifar-10? Of course, it can be easily pruned if the model is too much capacity for a simple dataset.  Why not try the Resnet-20 first?

[1] C. Louizos et al., Bayesian Compression for Deep Learning, NIPS, 2017
[2] J. Achterhold et al., Variational Network Quantization, ICLR, 2018

---

> ### Author Response · Authors · 2018-11-26
> **Reply to AnonReviewer4**
>
> We thank the reviewer for the feedback.
>
> 1) About "Ignoring the latest improvement in (C. Louizos et al., 2017) and (J. Achterhold et al.)":
>
> While we thank the reviewer for providing us more related works,  it worths noticing that pruning ratios in (C. Louizos et al., 2017), (J. Achterhold et al.) are not as strong as our compared baseline "Variational Dropout". For example, for LeNet on Mnist, the former have ~0.65%, while the latter (and our result) are less than 0.4%, and for VGG on CIFAR-10, the former have ~5.5%, while the latter (and our result) are less than 2%. That is, both our method and VD has better results compared to the two related works.
>
> Note many results provided in (C. Louizos et al., 2017), (J. Achterhold et al.) are for simultaneous pruning and quantization, while our submission focuses more on investigating the pruning effect of the simple L1 regularizer. In this work, we focus on the weight pruning ratio without quantization.
>
> (2) About  comment "Repeating the old story from other papers":
>
> Our story focuses more on the analysis of "problem" instead of the "algorithm". In other words, we argue that different problems have different compression rate, depending on their number of supporting labels, when a simple L1-regularized pruning objective is used.  The algorithm we proposed is just a tool for helping our iterates getting closer to the stationary points.
>
> (3) About comment "quite limited novelty":
>
> Firstly, our novelty lies more on the analysis of the pruning objective than on the algorithm. Second, it is a wrong impression that we are proposing ADAM over SGD.  Our proposition for the algorithm is the "L1 cumulative" technique as a general extension module to modify any stochastic-gradient-based algorithms, such as SGD and ADAM, into a sparsity-inducing solver.
>
> (4) About comment "lacking solid experiments":
>
> The sentence is an editorial mistake. We will strengthen our experiments in future revisions.

---

### Meta-Review · Area_Chair1 · 2018-12-16
**A study on sparse properties of L1-regularization in deep neural networks, yet experimental supports seem week.**

**Confidence:** 3
**Recommendation:** Reject

**Metareview:**

This paper studies the properties of L1 regularization for deep neural network. It contains some interesting results, e.g. the stationary point of an l1 regularized layer has bounded number of non-zero elements. On the other hand, the majority of reviewers has concerns on that experimental supports are weak and suggests rejection. Therefore, a final rejection is proposed.